# Investigating the role of AI explanations in lay individuals' comprehension of radiology reports: A metacognition lens

Yegin Genc[1]*, Mehmet Eren Ahsen[2,3], Zhan Zhang[1]

**1** Seidenberg School of Computer Science and Information Systems, Pace University, New York, New York, United States of America, **2** Department of Business Administration, University of Illinois at Urbana-Champaign, Champaign, Illinois, United States of America, **3** University of Illinois at Urbana-Champaign, Carle Illinois School of Medicine, Urbana, Illinois, United States of America

* ygenc@pace.edu

## Abstract

While there has been extensive research on techniques for explainable artificial intelligence (XAI) to enhance AI recommendations, the metacognitive processes in interacting with AI explanations remain underexplored. This study examines how AI explanations impact human decision-making by leveraging cognitive mechanisms that evaluate the accuracy of AI recommendations. We conducted a large-scale experiment (N = 4,302) on Amazon Mechanical Turk (AMT), where participants classified radiology reports as normal or abnormal. Participants were randomly assigned to three groups: a) no AI input (control group), b) AI prediction only, and c) AI prediction with explanation. Our results indicate that AI explanations enhanced task performance. Our results indicate that explanations are more effective when AI prediction confidence is high or users' self-confidence is low. We conclude by discussing the implications of our findings.

## 1. Introduction

As many AI systems are mostly working as a "black box", users face challenges in determining whether they can trust the AI recommendations and use that to make decisions [1]. When complex AI systems supplement decisions that play an essential role in human lives such as those in healthcare, understanding how the complex AI models generate their recommendations (i.e., model predictions) becomes critical to AI system users [2]. Therefore, there has been a push from the community to increase the focus on explainable AI (XAI) when humans and AI agents work together [3–5]. The rise of the XAI domain aims to promote trust and acceptance of complex AI models that often yield opaque predictions [3,6,7]. Prior work has demonstrated that providing explanations on the inner workings of AI systems can augment

**Data availability statement:** Data of the study will be available at https://github.com/ygenc/xai_metacognition/blob/main/mturk_responses.csv.

**Funding:** This work was supported in part by funding from the National Institute of Health (Award# 1R15LM014556-01).

**Competing interests:** The authors have declared that no competing interests exist.

decision making by providing system users further information from the model than a simple binary prediction to support decisions [8–10].

While the effect of explanations in mitigating the limitation of imperfect information provided by AI agents is significant, recent studies have revealed a higher-order relationship between humans and AI agents when together [11,12]. For example, recent studies have unveiled that human agents go through metacognitive processes when interacting with AI systems [11,12]. Metacognition involves the awareness and understanding of one's own cognitive processes [13]. These metacognitive processes manifest in two separate yet interconnected facets: system monitoring cognition (judgment calls by human agents based on their own preferences while assessing the validity of the AI predictions) [14] and self-monitoring cognition (the self-aware thought processes about their own decision-making strategies and reasoning, as they make decisions based on the AI system's outputs) [11,12,15]. XAI has predominantly been conceptualized within the scope of system monitoring cognition because explanations primarily aim to elucidate the accuracy of AI recommendations [16]. We argue that XAI should be considered within a broader metacognitive context that encompasses self-monitoring cognitions, for three primary reasons outlined below.

First, decision-makers have been found to follow different metacognitive patterns following AI advice that confirms and disconfirms their initial framing (i.e., initial intuition of the decision). Specifically, AI advice perceived as accurate (i.e., confirming the initial framing) is less likely to unfold metacognitive conflicts that trigger belief conformity and systems justification processes [17]. Therefore, decision-makers will likely engage with explanations differently when they perceive the AI recommendation as accurate vs. inaccurate.

Second, research on augmented decisions with AI found that the two metacognitive processes (system and self-monitoring) "can condition each other, and their dynamic interplay influences decision outcomes" [11]. For example, when decision-makers engage in active consideration, they compare their decision process and machine reasoning using the data elements that support AI advice. As a result, they may refine their perception of the accuracy of their original intuition and the perceived system accuracy. In this view, explanations that aim to support AI advice may influence the refinement of the confidence in the initial framing and the perceived system accuracy.

Finally, metacognitive monitoring constitutes a sensing activity that allows decision-makers to regulate the degree of deliberate, systematic reasoning (as opposed to quick, heuristic reasoning) and the amount of information sought [13,18]. Balancing systematic and heuristic reasoning influences how individuals cope with the cognitive challenges posed by potentially incorrect advice [19,20]. Explanations that reduce the cognitive challenges of detecting potentially inaccurate advice are likely to influence this balancing activity.

Driven by the research gap in investigating self-monitoring cognition in human-AI collaboration, we conducted an online experiment with 4,302 participants using the Mechanical Turk platform [21]. In this experiment, the participants were asked to identify whether the anatomical structure described in a radiology report excerpt

represents a normal or an abnormal diagnosis. During the task, the participants in experimental groups were also provided with an AI-based prediction of the anatomical structure. Participants were randomly assigned to conditions where explanations for the AI-based predictions were added or not. Our results validate the impact of explanations on the metacognitive assessment of the human agents' abilities, namely self-monitoring processes. Our results also show that human agents are more likely to correct an incorrect initial intuition (initial framing) when AI recommendations are explained, regardless of the accuracy of the AI recommendation. This suggests that even an incorrect AI recommendation, when explained, can help human agents recover from incorrect initial framing. Further, we find that the overall positive effects of explanations are more significant when AI prediction confidence is high and human self-confidence is low.

For instance, the applicability of explanations within AI systems extends beyond existing assumptions, notably trust and user acceptance. AI explanations can facilitate users' introspection and awareness of their cognitive biases. Furthermore, our findings hold significance in the realm of healthcare, highlighting the importance of AI explanations and self-monitoring frameworks in mitigating human errors. By diminishing the prevalence of such errors, the quality of care provided can be significantly enhanced, potentially resulting in more precise diagnoses and efficacious treatments.

The following sections provide the theoretical background for explainable AI and metacognition and describe the theoretical framework (Section 2). We then proceed to test the model experimentally (Section 3). Finally, we report major findings (Section 4) and conclude with a discussion of the implications of our results for theory and practice (Section 5).

## 2. Theoretical background and hypotheses

### 2.1. XAI: Explainable AI

As AI systems become increasingly complex and embedded in decision-making environments, the need to make their logic transparent and trustworthy has grown substantially [22–24]. This increased demand for transparency led to both technical studies that design and develop explainable components for AI (see Hassija et al. [7] and Arrieta et al. [25] for extensive reviews of current methods) and theoretical studies that aim to ground the technical work with theories of explainable AI [16,26–28]. Others have focused on reviewing developments in XAI research by creating taxonomies based on the techniques used to generate explanations and the scope of explanations [7,29–31]. Similar studies also reviewed explainable AI, particularly in the medical domain [32–35].

Recent studies emphasize that explanations are not only technical tools but also socio-cognitive mechanisms that influence trust, learning, and user engagement in human-AI systems [36,37]. Research also highlights the importance of designing for interpretability, particularly in dynamic digital contexts where algorithmic decisions are intertwined with user-generated content, such as ranking schemes [38]. These developments suggest a growing awareness that effective explainability depends on more than post-hoc transparency—it involves aligning system design with cognitive and organizational expectations [39,40].

From a technical perspective, explainability can be achieved by designing additional tools that enhance *prediction interpretation and justification* [41]. These tools commonly provide post-hoc explanations that clarify the inner functioning of complex models. These post-hoc explanations can be *local explanations* that focus on the less complicated solution subspaces that are relevant for the whole model [42,43]; *explanations by example* that extract data examples appropriate to the generated results [44–46]; *explanations by simplification* that are derived from a newly-trained model optimized for reducing predictive model complexity with minimal compromise from prediction accuracies [47,48]; and *feature relevance explanations* that aim to clarify the inner functioning of a model by computing a relevance score for its managed variables [49–51].

A second stream of technical research in explainable AI focuses on designing inherently more *interpretable models* [25]. Intelligent systems can be more understandable with the aid of knowledge bases during the design process [52]. Current approaches aim to align AI model features with the features of a knowledge base [53] that are simultaneously constructed during the model building [54]. Another approach to building an inherently interpretable method is to jointly

train a less complex but interpretable model with a more complex and accurate model in an ensemble-learning fashion [55,56]. The more transparent of the two models are then used to generate explanations.

While research on AI explanations has been growing, theoretical efforts in explainable AI often overlook the rich cognitive context in which humans interact with explanations in general [16,57]. Studies of human-AI interactions reveal the complex nature of cognitive processes, particularly metacognitive processes, when AI recommendations are involved in decision-making [11,12]. Understanding how these cognitive processes unfold when explanations are added to human-AI interactions can significantly enhance research in explainable AI. For example, recent studies highlight the importance of designing AI systems that can account for both ethical and cognitive concerns of end users, particularly in high-stakes environments [58], and emphasize how explanations can shape trust maintenance and restoration strategies, and long-term performance outcomes in AI-supported decisions [59,60].

### 2.2. Human metacognition in the research of human-computer interaction

Scholars have increasingly examined the higher-level cognitive processes that guide human judgment in interactions with technology [61,62], as these processes have been reported to be integral to the complex interplay between computing systems and information processing [63]. These higher-level cognitive processes, specifically metacognitions, monitor the progress of our decision-making activities, as well as the effort and time spent on these activities [13]. For example, meta-cognitions enable decision-makers to dynamically balance between quick, heuristic reasoning and deliberate, systematic reasoning [19,20]. In the context of human-computer interaction, these processes may improve our use of technology, e.g., improve team effectiveness in software development [64], or lead to systematic biases when using technology, e.g., confirmation bias in online reviews [65].

Recently, metacognitions have gained prominence in theorizing about the cognitive challenges associated with augmented decisions that involve AI advice [11,66,67]. With augmented decision-making involving the judgment calls that human agents make based on their preferences while assessing the validity of AI predictions (system-monitoring meta-cognitions) [11], the decision performance of human agents is also influenced by their assessment of their own abilities (self-monitoring metacognitions) [15]. Furthermore, current perspectives on the integration of humans and machines conceptualize these hybrids as sociotechnical systems, wherein both machines and human agents engage in collaborative learning within metahuman systems [12]. This perspective highlights the advanced learning capabilities linked to shifting goals and assumptions about the nature of learning itself, particularly when agents with differing cognitive architectures, namely machines and humans, are more closely integrated [12]. Seidel et al. [68] mention a feedback loop in this learning process where human agents learn about the "mental models" – i.e., decision models – embedded in the machine agents.

In addition, metacognitive error monitoring processes are also interrelated with the decision-makers' self-confidence [69] and the credibility [70] or persuasiveness [71] of the recommendations. For example, studies show both over and low self-confidence can adversely affect metacognitive processes [72,73]. Similarly, persuasive communications literature suggests that any message, such as AI explanations, can influence human cognition by affecting the content or the validity of thoughts [74]. The interaction between human metacognition and AI feedback is further complicated by the presentation and context of recommendations, as demonstrated in experiments involving confidence interfaces and trust calibration on digital platforms [e.g., 75].

These dynamics that surface when human and machine agents interact suggest a complex phenomenon with potentially multiple underlying mechanisms of human judgment and explanations about the mental models of machine agents. In this context, understanding how explanations that aim to support system-monitoring processes can also influence self-monitoring processes can provide further insights into the uses of explanations in human-AI interaction.

## 2.3. Theoretical framework and hypotheses development

As described earlier, more recent views on advice-taking suggest both heuristic and systematic reasoning are involved [13], and together they affect the cognitive processes that go beyond evaluating the machine advice (system-monitoring) and include evaluating decision-makers' own reasoning (self-monitoring) [11,12,15]. This new view focuses on the interplay between the intuitive assessment of the decision task (i.e., the initial framing) and the additional data provided in the form of machine advice as theorized in the Naturalistic Decision-Making (NDM) framework by Klein [76]. Following suit, we posit that just like the machine predictions, their explanations can also be involved in metacognitive processes that go beyond system-monitoring cognition because these explanations serve as additional data points about the task. Therefore, the interplay between the intuitive assessment of the task and the explanation of machine predictions, just like the prediction itself, is also relevant to the outcome of augmented decision-making. To study the said effects of explanations, we follow the NDM framework and focus on the metacognitive effects of explanations for augmented decision-making.

NDM suggests "[w]hen there are cues that a[n] [initial] judgment could be wrong, [the decision-maker replaces] intuition by careful reasoning" [76]. Therefore, we consider how explanations might serve as "additional cues" that suggest the intuitive judgment might be wrong. Particularly, we first consider the role explanations may play when machine predictions are in conflict with the initial framing. As Jussupow et al. [11] suggested, when machine predictions disconfirm the initial prediction, decision-makers need to overcome the "conflict between their beliefs in their own competence and their beliefs in the AI capabilities." And when explanations are present, they will likely influence the decision-makers' beliefs about the AI capabilities. Second, explanations can also challenge the initial framing if they reveal issues with the machine predictions that confirm the initial framing. Particularly, when explanations reveal issues with the machine algorithms that are hard to catch otherwise [23], decision-makers are likely to consider that the agreeing initial framing might also be wrong. This, as the NDM suggests, is likely to trigger careful reasoning regarding the task [76].

In our study, we explore the role of explanations in the context of the accuracy of the machine predictions, because the explanations will support the machine predictions or reveal the irregularities in the machine reasoning, depending on whether the machine is providing a correct prediction or not. While the technical efficiency or fidelity of the explanation mechanism itself (i.e., how accurately the explanation reflects the internal reasoning of the model) is also important, we assume that the explanation mechanisms used in the study provide sufficiently interpretable and reasonable outputs to be meaningful to participants. We consider system-level comparisons between explanation methods (e.g., SHAP, LIME, rule-based) beyond the scope of this study, which focuses instead on the behavioral and metacognitive effects of explanations as perceived by human users. Subsequently, to consider the conflicts with the initial framing, we also consider the accuracy of the initial framing as well.

Finally, considering the persuasive communications literature, the persuasiveness of the AI predictions and their explanations are likely to regulate their impact on the decision makers. Drawing from the literature, we examine two variables—self-confidence (i.e., quality of thoughts) and prediction confidence (i.e., the source credibility) [77]—to study influence of the persuasiveness of explanations. Against this backdrop, we develop our formal hypotheses within our framework, as described below.

**Self-monitoring hypotheses.** When machine predictions trigger cognitive conflicts by contradicting the initial framing (i.e., making recommendations that contradict with what the decision maker's intuition,) we posit that explanations can serve as additional support to validate the machine predictions [25]. The extent to which explanations support machines' contradictory predictions will influence the decision-makers' tendency to consider their initial framing and accept the machine predictions [11]. Simply put, conflicting predictions with explanations that make a stronger case for the projections can be more influential than the predictions without explanations. As the self-monitoring processes compare their incorrect initial framing to the conflicting yet correct AI recommendations, they are more likely to switch from their initial framing because the contradicting AI recommendation is strengthened by the explanation (i.e., *strong evidence*

*effect.*) However, the strengthening effect of explanation is less likely to appear when the AI recommendation is incorrect because explanations of the incorrect AI recommendations are likely to reveal the weaknesses of the predictive model. Therefore, the explanations are less likely to support the recommendations when the conflict is between a correct initial framing and an incorrect AI recommendation. Consequently, we expect this *strong evidence effect* of explanations to be relevant when the cognitive conflict is between an incorrect initial framing and a correct AI recommendation.

> **H1A:** *When initial framing is incorrect, and AI recommendations are correct, explanations help final decisions move towards the accurate AI recommendations and thus away from the initial framing.*

Explanations might also help users realize that their initial framing is incorrect when they raise doubt about the validity of the machine predictions that agree with the initial framing. Explanations that show machine reasoning weaknesses undermine the prediction outcome's validity. And the decision-maker, whose initial framing agrees with the machine predictions, may reconsider the validity of their own reasoning after recognizing the weakness in the machine reasoning. A similar phenomenon has been noted in "weak argument" literature, where when an argument is supported by weak evidence, it is less likely to gain support than when no evidence is presented [78]. Fernbach et al. [78] argue this "weak evidence effect" arises in part because alternative causes are weighted more when humans are reasoning for diagnosis rather than making predictions [79]. Accordingly, as the self-monitoring processes compare the incorrect initial framing to the confirming (yet inaccurate) AI recommendations, they are less likely to stay with their initial framing because the confirming AI recommendations are weakened by the explanation. However, this weakening effect is less likely to appear when the initial framing agrees with the correct AI recommendations because explanations are likely to strengthen the position of the AI recommendations when the predictions are accurate. Therefore, when the agreement is between a correct initial framing and a correct AI recommendation, the explanations are less likely to weaken the recommendation. Consequently, we expect this *weakening evidence effect* to be relevant when both the initial framing and the AI recommendation are incorrect.

> **H1B:** *When both initial framing and AI recommendations are incorrect, explanations help final decisions move away from the incorrect AI recommendations and thus away from the incorrect initial framing.*

As discussed above, the possible effects of explanations may differ based on the accuracy of the initial framing by the decision-maker and the machine prediction. Explanations serve as "weak evidence" against confirming inaccurate AI as they reveal the inaccuracies of the machine predictions and "strong evidence" against disconfirming accurate AI as they support the accurate machine predictions. Taken together, we see that the positive effect of explanations is more prevalent when the initial framing is incorrect.

> **H1:** *When controlled for recommendation accuracy, the positive impact of explanations will be more prevalent when the initial framing is incorrect.*

Table 1 below summarizes our framework in terms of the impact of explanations in the context of the accuracy of the initial framing and the machine predictions.

**Persuasiveness hypotheses.** AI explanations primarily aim to persuade that the predictions derived from machine-learning-based algorithms are justifiable [80]. The literature on persuasive communications suggests that any message, such as AI explanations, can influence human cognition by affecting the content or the validity of thoughts [74] and the quality and source credibility of these messages will impact how persuasive these messages will be [77]. The source for explanations are typically the predictive models they aim to explain since explanations are generated by isolating the contributions of individual features of the predictions [25]. We can reason that the persuasiveness of an AI explanation will be related to the credibility of the predictions they explain. Therefore, we can expect the explanations of predictions with high prediction confidence to be more persuasive.

> **H2:** *The positive impact of explanations is more substantial when the prediction confidence of the machine is high.*

Further, based on the well-established relationship between attitude confidence and associated behavior [81], we expect confidence in one's thoughts to influence how they will utilize external information (i.e., the AI explanations) in their

**Table 1. Conceptualizing the effect of explanations based on the initial framing and the machine prediction accuracies.**

| | Correct AI recommendations | Incorrect AI recommendations |
|---|---|---|
| | **Explanation Effects** | |
| | Strong evidence effect: Explanations of correct AI pull decision-makers towards the recommendations. | Weak evidence effect: Explanations of incorrect AI push decision-makers away from the recommendations. |
| **Incorrect Initial Framing (H1)** | With self-monitoring, moving towards recommendations may trigger moving away from initial incorrect framing. (H1A) | With self-monitoring, moving away from recommendations may also trigger moving away from initial incorrect framing. (H1B) |
| **Correct Initial Framing** | No effect – since both initial framing and the recommendations are correct. | No effect – Moving away from recommendations does not trigger moving away from the initial correct framing since it was already contradicting with incorrect recommendation. |

decision-making. If decision-makers have lower self-confidence about their intuition (i.e., initial framing), they are more likely to seek further information or additional cues, i.e., explanations [82]; and, vice versa [83,84]. Therefore, we expect to observe the positive effects of explanations in cases where humans are less confident about their judgment because they are more likely to utilize explanations.

*H3: The positive impact of explanations is more substantial when self-confidence in human judgment is low.*

In summary, our study aims to investigate the impact of AI explanations and confidence on human decision through metacognitive processes. Fig 1 summarizes our conceptual framework.

## 3. Methods

### 3.1. Experimental design

Following the calls for using online experimentation to study behavioral aspect of technology in the information systems field [85], we conduct an online experiment to test our hypotheses. Participant recruitment and data collection took place between 04/15/2019 and 05/24/2019. Participants were presented with an online consent form and provided consent by clicking on a button that read "I certify that I read and understand the informed consent". Their consent is recorded in the dataset. The experiment involves identifying whether the anatomical structure described in a radiology report excerpt is

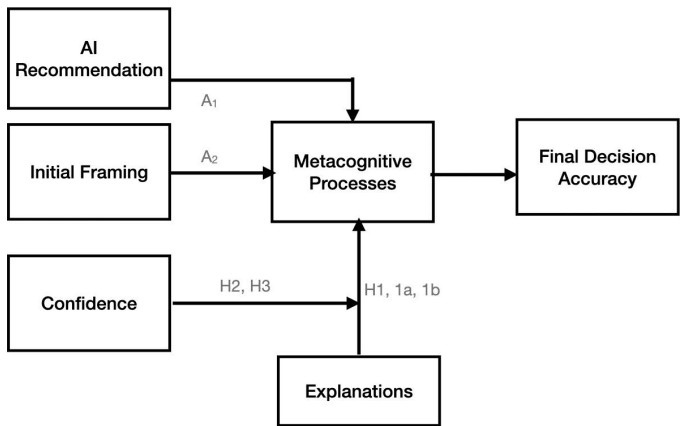

**Fig 1. The theoretical model.** $A_1$ and $A_2$ are adopted from Jussupow et al. (2021). This study investigates the impact of explanations (H1) and confidence (H2, H3) on the final decision accuracy in the context of previously identified metacognitive processes.

abnormal. The report excerpts are from the Audiological and Genetic Database (https://audgendb.github.io). For example, we ask participants to label the normality of the following sentences: "No evidence of erosion in the incus and malleus." To provide context for the sentence, we also provide the sentences preceding and following from the original report.

Our task aims to examine the effects of AI predictions on human decisions and how explanations can modulate such effects. To that end, we create treatment group based on the presence of an explanation for the predictions (Explained vs. Not Explained). We select a control group where the participants do not see an AI prediction during the prediction task. As a result, participants are randomly assigned to one of the three conditions: 1) AI predictions with explanations; 2) AI predictions without any explanations; 3) without any AI prediction (control group). An example of a task under each condition is shown in Fig 2.

### 3.2. Building an AI recommendation model

**Data.** The dataset prepared by Cocos et al. [86] had 276 sentences that were previously labeled by the subject matter experts and 727 sentences that were crowdsourced to be labeled. We use the expert labels as the "gold standard" data to evaluate the accuracy of the AI predictions and participants' decision outcomes. We use the 727 crowd-annotated data set to train the two AI models that provide the predictions for the experiments. Using the crowd-annotated dataset, we train two predictive models using Least Squares Support Vector Machine Classifiers with linear kernels.

**Recommendation models.** To build AI-based recommendations, we train Support Vector Machine (SVM)-based predictive models. SVM performs better than intrinsically transparent models and, similar to the state-of-the-art deep learning algorithms, requires further explanations that are created post-hoc (see Arrieta et al. [25] for a detailed classification of models based on their transparency). SVMs keep a fair balance between performance and interpretability and have been applied in various fields [87–89]. We avoid complex models such as deep neural networks due to the limited training data size [90]. In particular, we created two models: a "high-accuracy model" trained on the full set of 727 samples, and a "low-accuracy model" trained on a randomly selected 20% subset of the same data. The performance of the two models was evaluated using the accuracy metric on the experimental dataset containing 276 expert-labeled sentences, with classification accuracy calculated as the proportion of correct predictions, given by the formula:

$$Accuracy = \frac{Number\ of\ Correct\ Predictions}{Total\ number\ of\ prediction}$$

Based on this measure, the "low-accuracy model" achieved an accuracy of 0.702, while the "high-accuracy model" achieved a substantially higher accuracy of 0.891, confirming the intended disparity in predictive performance. To assess

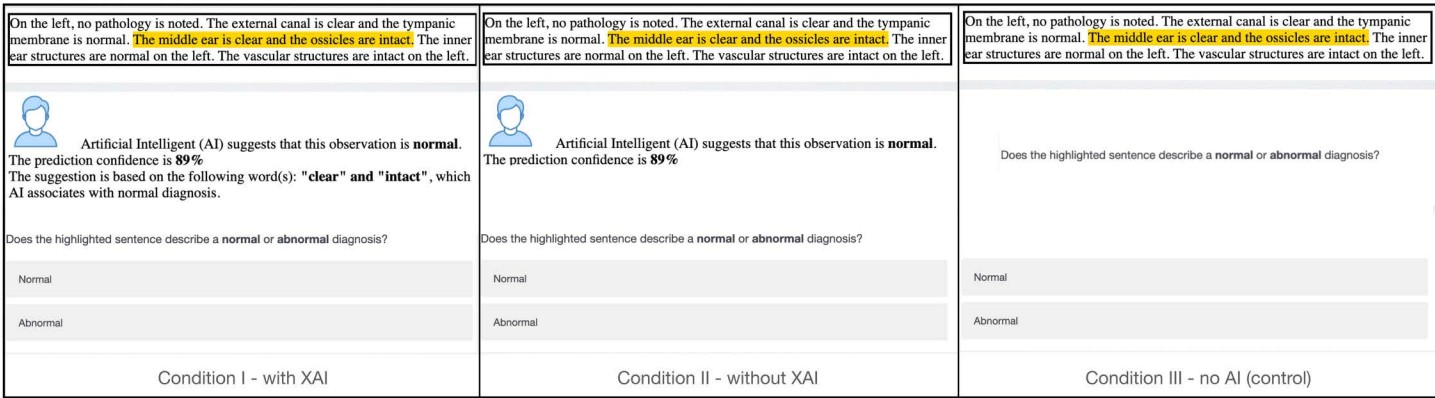

**Fig 2. An example of the decision task used in the online experiment in three conditions.**

the statistical significance of this difference, we conducted DeLong's test for two correlated ROC curves, which yielded a p-value of $1.83 \times 10^{-10}$, indicating a highly significant difference between the two models.

Following prior work (e.g., [28,42,69]), we generated local explanations for each individual prediction made by the models, using the corresponding input features as the basis for these explanations. These local explanations aim to provide interpretable insights into how specific features influenced a given prediction, thereby offering a more transparent view of model behavior at the instance level. By focusing on the feature contributions for each prediction, we were able to assess the model's decision-making process in a nuanced and context-specific manner. The next section provides a detailed overview of the methodology used to construct these explanations, including the tools, algorithms, and interpretability techniques employed.

**Explanations.** Our annotation tasks are sentences; we need to construct features from them to build a machine-learning model. We extract 1-grams and bigrams from the annotation tasks, a popular way that has been frequently used in the natural language processing literature [91]. We use the Python Scikit-learn package for creating the SVM [92]. The training of the SVM model results in a ranking of words or bigrams (features) in the order of their significance. We use this ranked list of features to construct explanations. As a result of this, tasks in the explained condition include an additional sentence detailing what part of the to-be-annotated sentence is most significantly associated with the AI prediction. For example, for the following sentence to be annotated: "*The external canal is clear, and the tympanic membrane is normal*," the AI model predicts (correctly) that the observation should be annotated as "normal." In the explained condition, we include the following text: "*The suggestion is based on the following word(s): 'normal' and 'clear,' which AI associates with a normal diagnosis.*" In this case, the underlined words, *normal* and *clear*, are the most significant features in the annotation prediction of the AI model. The difference between explained and not-explained conditions can be seen in Fig 2.

### 3.3. Participants and data collection process

We conduct our experiments on AMT, an online crowdsourcing platform that facilitates worker recruiting, human intelligence task (HIT) completion, and payment. To partially account for individual differences in cognitive and technical aptitude, we applied strict participant quality filters and included self-reported measures of technical literacy and AI literacy as control variables in our analysis, helping to mitigate variance related to familiarity with technology and decision-support systems. In particular, we invite English-speaking MTurk workers who reside in the United States and who have completed at least 1,000 previous HITs with at least a 95% approval rate to participate in our task. We collect annotations in batches, which are deployed in sequence. Each batch contains tasks designed for one condition. We populate the HITs for each batch by randomly selecting a sentence from the current collection of unlabeled sentences. Table 2 summarizes the demographic characteristics of all participants. We embed two quality assurance mechanisms in each annotation task to minimize the impact of the varying expertise of crowd workers. First, we require three unique workers to label each sentence. Second, workers can only complete one task to avoid any carryover effects.

Participant recruitment and data collection took place between 04/15/2019 and 05/24/2019. In the beginning, participants filled out an online consent form, acknowledging they read the consent form and understood the demands of participating in the study. Upon giving consent, participants complete a demographics questionnaire on a survey-like interface where we provide instructions and examples to help the participants get familiar with the annotation task. Following instructions, we ask participants to classify the highlighted sentences as describing a normal or an abnormal observation of the specified component (Fig 2). Toward the end of the study, participants are asked to complete a survey indicating their domain expertise, trust, and knowledge of AI technology, etc. (as described in Section 3.4.5).

### 3.4. Measurements and variables

**Accuracy measures.** Our accuracy evaluations for both the human and AI are based on the expert responses that serve as the gold standard. For each task, we collected responses from at least three participants and determined

**Table 2. Summary of participants' demographics.**

| | | Control | With Explanation | Without Explanation |
|---|---|---|---|---|
| Gender | Female | 52% (458) | 55% (926) | 55% (953) |
| | Male | 47% (417) | 44% (747) | 45% (779) |
| | Other | 1% (7) | <1% (7) | <1% (8) |
| Age | 18-25 years | 20% (175) | 17% (279) | 20% (343) |
| | 26-49 years | 65% (572) | 67% (1125) | 65% (1136) |
| | 50-64 years | 12% (109) | 13% (220) | 12% (215) |
| | 65 and older | 3% (26) | 3% (56) | 3% (46) |
| Occupation Industry | Education | 10% (85) | 12% (206) | 13% (232) |
| | Finance | 11% (95) | 8% (132) | 10% (173) |
| | Government | 5% (42) | 5% (89) | 5% (94) |
| | Health Care | 17% (147) | 17% (293) | 19% (337) |
| | IT | 21% (181) | 19% (326) | 18% (313) |
| | Other | 38% (332) | 38% (634) | 34% (591) |

task-level accuracy by comparing the aggregated responses (i.e., determined by majority vote) to the gold standard expert responses. We define *initial framing accuracy* as the majority vote accuracy in the control condition, where participants completed the task without access to an AI prediction. In contrast, *final decision accuracy* refers to the majority vote accuracy for the same task when participants were shown an AI prediction.

**Explanation effect.** The presence of an explanation is one of the treatments in our experimental setup. In "explained" conditions, we provide an explanatory sentence that suggests what part of the to-be-annotated sentence is associated with the AI prediction. Our explained condition measure is 1 when an explanation is present and 0 otherwise.

**Machine confidence.** Machine confidence is defined as the AI model's confidence level, computed as a function of the posterior probabilities of individual predictions and the model's overall accuracy to capture both instance-level certainty and overall reliability.

**Self-confidence.** We measure participant self-confidence for each task based on their confidence in their answers when AI predictions aren't available. For each task, we measure the confidence among the three participants under the control condition (since participants did not see any AI prediction in this condition) and calculate the average self-confidence score for the task. We ask participants to self-report confidence in their answers. They rate their responses on the Likert Scale from 1, not at all confident, to 5, very confident (mean = 4.00, sd = 0.97).

**Control variables.** To control for the model quality, in terms of its overall accuracy, we build several SVM models by varying the size of the training set. For each SVM model, we use five-fold cross-validation to fine-tune its parameters. During the experiments, participants were randomly assigned to tasks that showed AI predictions either from a model that used all the training data (727 samples) as the "high-accuracy model" or the one that used 20% of the training data as the "low-accuracy model." Each task was completed both with high-accuracy model predictions and low-accuracy model predictions by different participants.

To control for factors that can influence participants' judgment accuracy other than model accuracy, we collect more data from them during the experiment.

Domain expertise has been discussed to influence how individuals utilize explanations when algorithmic decisions are transparent. For example, Dhaliwal et al. [52] argued that expertise influences how explanations provided by knowledge-based systems are utilized. Specifically, they suggest novice users can make greater use of explanations. Also, the overall benefits of these explanations for expert and novice users can vary depending on the provisioning strategy. In essence, novice users can benefit more from the explanations preceding predictions (feedforward), while experts

can benefit more from the explanations following predictions (feedback). To that end, we control for the participants' *domain expertise*. We ask participants to self-report healthcare literacy. They rate their responses on the Likert Scale from 1, not at all, to 5, very well (mean = 3.95, sd = 0.85). Second, we consider the participants' trust in AI predictions. Trust in technological artifacts influences their use [93]. From a decision perspective, trust is also an essential factor in delegating a task to "intelligent agents," and the interfaces through which human agents intact with these intelligence agents can and should increase trust [94,95]. Moreover, trust also serves as a proxy for participants' a priori assumptions about the correctness or reliability of AI predictions—a form of baseline framing that may influence how individuals interpret and integrate AI recommendations.

Additionally, Gregor et al. [96] suggested that transparent agents that provide explanations can improve performance because users would feel more comfortable if the intelligent agent could explain its actions. Overall, these perspectives suggest that trust in machine-generated advice is a complex concept [97,98] and will affect the interaction between human and AI agents. To that end, we control for the participants' trust in AI. We ask participants to self-report their trust in Artificial Intelligence. They rate their responses on the Likert Scale from 1, not at all, to 5, very well (mean = 3.22, sd = 0.97).

Finally, we consider the participants' knowledge of AI technology. It has been reported that one way to improve the proper utilization of algorithmic aids is to increase algorithmic literacy among human judges. That is, if human judges are more knowledgeable about interpreting statistical outputs such as decision accuracy and appreciate the utility of decision aids under certainty, their interaction with these tools will be positively influenced [99,100]). To that end, we control for the participants' knowledge about Artificial Intelligence. We ask participants to self-report their knowledge about Artificial Intelligence before the task. They rate their responses on the Likert Scale from 1, not at all, to 5, very well (mean = 3.25s, sd = 0.95). The means and correlations of all variables appear in Table 3.

## 4. Results

### 4.1. Effects of explanations

To examine the effects of explanations, we perform a multivariate logistic regression analysis. With final decision accuracy—defined as a binary variable equal to 1 if the majority-vote response matches the expert-provided gold standard and 0 otherwise—as the dependent variable, we introduce independent variables in a stepwise fashion to account for potential instability arising from multicollinearity. The results are shown in Table 4, with Final Judgment Accuracy as the dependent variable. Model 1 includes these control variables: *Technical Literacy, Domain Expertise, AI Literacy, Trust to AI,* and *Model* (high vs. low accuracy model). In Model 2, we examine the effect of the *correct AI prediction*. The correct

**Table 3. Means and correlations for main hypotheses dependent variables.**

| Variable | M | SD | 1 | 2 | 3 | 4 | 5 | 6 | 7 | VIF |
|---|---|---|---|---|---|---|---|---|---|---|
| 1 Technical Literacy | 4.15 | 0.43 | | | | | | | | 1.30 |
| 2 Domain Expertise | 3.96 | 0.48 | .32** | | | | | | | 1.17 |
| 3 AI Literacy | 3.25 | 0.56 | .40** | .23** | | | | | | 1.41 |
| 4 Trust to AI | 3.21 | 0.54 | .18** | .07* | .37** | | | | | 1.22 |
| 5 Model Complexity | 0.50 | 0.50 | 0.05 | 0.01 | −0.01 | 0.03 | | | | 1.14 |
| 6 Correct Initial Judgment | 0.82 | 0.38 | −0.01 | .08** | 0.01 | −0.01 | 0.00 | | | 1.01 |
| 7 Correct AI | 0.80 | 0.40 | −0.04 | −.06* | −0.01 | 0.04 | −.23** | .09** | | 1.94 |
| 8 Transparency Effect | 0.30 | 0.64 | −0.03 | 0.01 | −0.01 | 0.03 | −.13** | 0.04 | .58** | 3.75 |

Notes: * indicates $p < .05$. ** indicates $p < .01$.

**Table 4. Results for regression analysis for final decision accuracy with AI recommendations.**

| | | DV: Final Decision Accuracy | | |
| --- | --- | --- | --- | --- |
| | | **Model 1** | **Model 2** | **Model 3** |
| CONTROLS | Technical Literacy | 0.119 (0.236) | 0.236 (0.263) | 0.209 (0.265) |
| | Domain Expertise | 0.451** (0.201) | 0.626*** (0.216) | 0.643*** (0.217) |
| | AI Literacy | −0.137 (0.191) | −0.149 (0.207) | −0.159 (0.209) |
| | Trust to AI | −0.073 (0.183) | −0.208 (0.199) | −0.178 (0.199) |
| | Model | 0.347* (0.183) | −0.198 (0.209) | −0.202 (0.210) |
| | Correct AI Prediction | | 2.304*** (0.211) | 2.320*** (0.212) |
| | Explanation | | | 0.431** (0.200) |
| | Constant | 0.098 (1.070) | −1.874 (1.193) | −2.099* (1.210) |
| | Observations | 1048 | 1048 | 1048 |
| | Log Likelihood | −412.933 | −349.952 | −347.591 |
| | Akaike Inf. Crit. | 837.866 | 713.904 | 711.183 |
| | **Adjusted pseudo R²** | 0.02 | 0.22 | 0.23 |

Note: *$p < 0.1$; **$p < 0.05$; ***$p < 0.01$

AI prediction has a significant positive relationship with the final decision accuracy, suggesting the final decision is more likely to be accurate when decision-makers are presented with a correct AI prediction rather than an incorrect AI prediction. Model 3 includes the *explanation effect*, suggesting that the presence of explanations will influence the final decision accuracy. The presence of explanation is strongly and positively associated with final decision accuracy, indicating that decisions are more likely to align with the gold standard when an explanation is provided.

### 4.2. Self-monitoring hypotheses (H1, H1A and H1B)

To test the differential effect of initial framing (H1) on the relationship between explanations and final decision accuracy, we conduct a sample-split analysis. We split our data depending on whether the initial framing for each task is accurate. We measure the initial framing accuracy based on the majority vote decision for the task when an AI prediction is not prompted. The results of our split sample analyses are shown in Table 5. The results based on this classification are shown in Model 3 and Model 4 in Table 5. Our results show that the impact of explanations is greater and significant when the initial framing is incorrect, supporting H1.

**The effects of explanations on incorrect initial framing (H1A and H1B).** To test the influence of explanations on incorrect initial framing with respect to the AI accuracy, we compare the final decision accuracy with and without the explanations particularly for the tasks where initial framing was captured as incorrect based on the answers without AI recommendations (i.e., control group answers). First, we examine how explanations affect participants' initial incorrect framing when AI predictions are correct. To do this, we use two-sample proportion tests to compare the accuracy rates (i.e., the proportion of correct responses) between conditions with and without explanations. Among the cases where initial

**Table 5. Results for subsample analyses for final judgment accuracy that agree with AI recommendations.**

| | | DV: Final Decision Accuracy | | | | | | |
|---|---|---|---|---|---|---|---|---|
| | | Model 3 | Model 4 | Model 5 | Model 6 | Model 7 | Model 8 | Model 9 |
| | | Overall | Initial Framing (H1) | | Machine-Confidence (H2) | | Self-Confidence (H3) | |
| | | | Incorrect | Correct | Low | High | Low | High |
| CONTROLS | Technical Literacy | 0.209 (0.265) | 0.404 (0.507) | 0.093 (0.327) | −0.192 (0.344) | 0.906** (0.435) | 0.029 (0.375) | 0.322 (0.378) |
| | Domain Expertise | 0.643*** (0.217) | 0.421 (0.427) | 0.603** (0.267) | 0.804*** (0.281) | 0.359 (0.348) | 0.689** (0.280) | 0.468 (0.335) |
| | AI Literacy | −0.159 (0.209) | −0.132 (0.420) | −0.133 (0.251) | −0.091 (0.271) | −0.188 (0.333) | 0.013 (0.281) | −0.256 (0.307) |
| | Trust to AI | −0.178 (0.199) | 0.102 (0.379) | −0.267 (0.244) | −0.275 (0.249) | −0.070 (0.332) | −0.173 (0.264) | −0.256 (0.301) |
| | Model | −0.202 (0.210) | −0.297 (0.400) | −0.121 (0.256) | −0.305 (0.263) | −0.048 (0.352) | −0.107 (0.423) | −0.305 (0.405) |
| | Correct AI Prediction | 2.320*** (0.212) | 2.319*** (0.423) | 2.281*** (0.258) | 2.288*** (0.270) | 2.442*** (0.352) | 1.916*** (0.274) | 2.895*** (0.332) |
| | Explanation | **0.431** (0.200)** | **0.830** (0.393)** | 0.314 (0.242) | **0.686*** (0.254)** | 0.053 (0.325) | **0.649** (0.267)** | 0.219 (0.295) |
| | Constant | −2.099* (1.210) | −4.048** (2.043) | −0.933 (1.569) | −1.332 (1.571) | −3.566* (1.924) | −2.052 (1.712) | −1.555 (1.738) |
| | Observations | 1,048 | 184 | 864 | 555 | 535 | 478 | 612 |
| | Log Likelihood | −347.591 | −88.875 | −243.928 | −215.3 | −134.807 | −190.349 | −167.457 |
| | Akaike Inf. Crit. | 711.183 | 193.751 | 503.855 | 446.599 | 285.614 | 396.698 | 350.914 |
| | **Adjusted pseudo R²** | 0.23 | 0.3 | 0.2 | 0.25 | 0.22 | 0.21 | 0.25 |

Note: *$p < 0.1$; **$p < 0.05$; ***$p < 0.01$

framing is incorrect and AI predictions are correct, the accuracy of final judgment is significantly higher when AI predictions are with explanations ($p_{explanation} = 88.9\%$) than without explanations ($p_{no\_explanation} = 76.4\%$) ($H_A$: $p_{explanation} > p_{no\_explanation}$, $p < 0.05$). The significant difference suggests that when decision-makers with incorrect initial framing are prompted with correct AI predictions, the use of explanations leads to more accurate decisions, supporting H1A. The effect of explanations under this condition is also shown in Fig 3 in the left column.

Second, we look at the effect of explanations on incorrect initial framing when AI predictions are also incorrect. With a two-sample proportion test, we compared the accuracy rates (i.e., the proportion of correct responses) between conditions with and without explanations. Among the cases where both initial framing and AI predictions are incorrect, the final decision accuracy is significantly higher with explanations ($p_{explanation} = 50\%$) than without explanations ($p_{no\_explanation} = 25\%$.) The difference is significant ($H_A$: $p_{explanation} < p_{no\_explanation}$, $p < 0.05$). This result suggests that when decision-makers with incorrect initial framing are prompted with incorrect AI predictions, the use of explanations leads to more accurate decisions, supporting H1B. The effect of explanations under this condition is also shown in Fig 3 on the right column.

To assure that explanations are significantly effective when initial framing is incorrect, we conducted a subsample analysis by splitting the data based on whether the initial framing for the questions were correct or not. The results are shown in Model 4 and Model 5 of Table 5. The results suggest that the effect of explanations is significant (and positive) when the initial framing is incorrect.

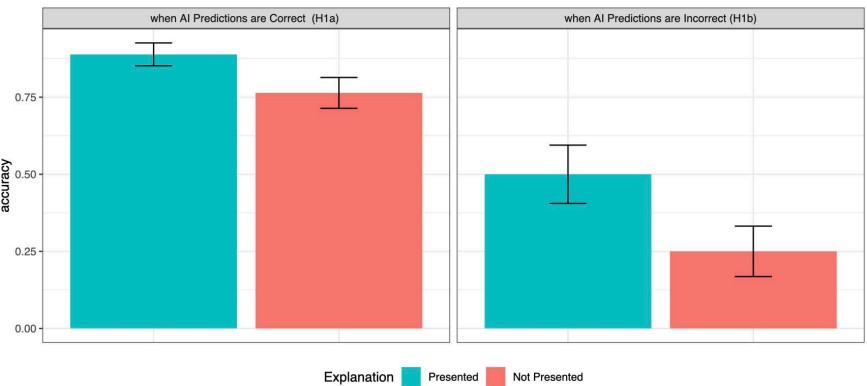

The Review of Explanation Effect for when the initial framing is incorrect

**Fig 3. The final decision accuracy rates when initial framing is incorrect.** The graph is categorized based on whether the AI recommendations made available to the subjects are correct (left column) or not (right column.) Error bars indicate 1 standard error.

**Table 6. Summary of findings.**

| Hypothesis | | Method | Findings | Effect Size and Practical Significance |
|---|---|---|---|---|
| H1 | Explanations influence the final decision accuracy with incorrect initial framing. | Sample split analysis | Supported (p < 0.05) | Correct outcome is 2.3 times more likely with explanations when initial framing is incorrect. (Odds ratio = 2.3) |
| H1A | Explanations positively influence the final decision accuracy when AI predictions are correct. | Proportion Test | Supported (p < 0.05) | Explanation effect size can be considered as moderate when predictions are correct (Cohen's h = 0.52). |
| H1B | Explanations positively influence the final decision accuracy when AI predictions are incorrect. | Proportion Test | Supported (p < 0.05) | Explanation effect size can be considered as small to medium when predictions are incorrect (Cohen's h = 0.34). |
| H2 | Explanations are more effective if AI prediction confidence is high. | Sample split analysis | Supported (p < 0.01) | Correct outcome is 2 times more likely with explanations when AI prediction confidence is high. (Odds ratio = 1.99) |
| H3 | Explanations are more effective if human's self-confidence is low. | Sample split analysis | Supported (p < 0.05) | Correct outcome is 1,9 times more likely with explanations when human's self-confidence is low. (Odds ratio = 1.91) |

### 4.3. Persuasiveness hypotheses (H2 and H3)

To study the effects of persuasiveness, we conduct two subsample analyses examining how different levels of machine confidence (H2) and self-confidence (H3) change the explanation effects. We continue to show the results of our split sample analyses in Table 5. We first split our data depending on whether the prediction probability is above the median prediction probability in the dataset. The results of this split are shown in Model 6 and Model 7 of Table 5. Our results show that the explanation effect is greater and significant with lower machine prediction accuracy, supporting H2. Similarly, we split the data based on whether the self-confidence score is above the median confidence score, and the results are shown in Model 8 and Model 9 of Table 5. Our results show that the effect of explanations is greater and significant with lower human self-confidence scores, supporting H3. Finally, a summary of our hypotheses, test methods, results and their practical significances are shown in Table 6.

### 4.4. Robustness tests

We conduct a series of additional analyses to examine the robustness of our results. First, our sample split analyses for machine confidence and self-confidence are based on the median values for confidence scores. To ensure that the results reported are robust, we replicate our analysis by comparing different sample splits. We conduct the same analysis

comparing 1) the first quartile with the fourth and 2) the first quartile with the rest, for both machine confidence and self-confidence scores. The results are similar to those of our original analysis.

Second, variable coefficients remain stable when added in a stepwise fashion, suggesting multicollinearity is not an issue. We further test for possible multicollinearity issues by computing the variance inflation factors (VIFs) for the exploratory variables. The VIF values for variables in the complete model, including the interaction term, are shown in Table 3. The VIF values in our model are less than the generally accepted threshold of 10 [101,102]. They are even lower than the more conservative suggested thresholds (i.e., the lowest VIF threshold we found was proposed by Pan et al., [103] as 4).

## 5. Discussion and conclusion

### 5.1. Theoretical contributions

Our study aims to expand the theories of XAI, which have drawn attention in the current literature [3,4,104]. Specifically, we consider the concerns raised by Miller [16] on how the theory efforts in XAI are missing the rich cognitive context when humans interact with explanations in general. This rich cognitive context is currently being recognized in the studies of human-AI interactions (e.g., [11,105]). Our work contributes to the XAI literature by studying the interactions of humans, AI, and their explanations through a theoretical lens that emphasizes this cognitive context.

Our findings suggest that explanations of AI decisions improve human-AI collaboration by overcoming the human's cognitive limitations when they trigger metacognitive self-monitoring processes. This view builds on Jussupow et al.'s [11] decision augmentation with AI, where they studied the role of human actors in compensating for technical errors. In our work, we integrate explanations into the hybrid. We show that explanations can mitigate human judgment errors not only when AI recommendations are correct (H1A) but also when recommendations are incorrect (H1B) through self-monitoring processes. Our findings show that, contrary to Jussupow et al.'s [11] assumptions, at least in some cases, even though both AI and human agents are incorrect, explanations may be triggering self-monitoring processes and lead to correct decisions. For instance, when AI recommendations are incorrect, participants often arrive at the correct final judgment if those recommendations are accompanied by explanations that appear unrelated to the diagnosis (e.g., referencing terms like "internal auditory," "canal," "left," or "cochlear nerve" as indicators of an abnormal diagnosis). This occurs even when the initial framings of these tasks were incorrect.

Our findings also show that the influence of explanations on the final decision accuracy depends on factors that impact how persuasive these explanations are perceived. In particular, we confirm that machine prediction confidence acts as an indicator of how persuasive the explanations are (H2) and decision makers' self-confidence impacts the effects of explanations (H3).

Our findings also align with Fügener et al. [15], suggesting that metacognitive processes can positively influence human-AI collaboration by addressing human agents' cognitive limitations. Their findings show that supporting humans' awareness of these cognitive processes (increasing metacognitive knowledge) can improve collaboration. Our findings expand this by offering that supporting human agents' use of cognitive processes with help of additional information, such as explanations, can also enhance collaboration.

From a broader perspective, our findings expand the current understanding of machine outcomes' role in the human agents' cognitive loop [68]. Lyytinen et al. [12] posit the emergent human-machine hybrids as sociotechnical systems where machines and humans learn jointly. However, the dynamics that surface when combining different cognitive architectures of humans and machines are still unclear [12]. Our findings provide evidence that understandability of the machine predictions triggered by explanations can influence this dynamic.

### 5.2. Design and practical implications

From a practical perspective, our results suggest that designers and developers of explainable AI systems should extend their focus beyond the effects of explanations on users' rational decision-making processes. Specifically, their impact on the effects of AI predictions has several important implications. First, explanation performance metrics may need to be

adjusted for their metacognitive-related effects. Our findings suggest that explanations can go beyond securing user trust or helping users make sense of machine predictions; they allow users to reflect on their own decisions. For example, for tasks where both human predictions (without an AI input) and AI predictions were inaccurate, human judgment shifted to correct prediction when they saw the explanations for the incorrect AI predictions.

Second, our results suggest that when human decisions are involved, explanations can improve the final decision accuracy even when the models are trained on small datasets. This finding is particularly significant for AI implementations where the training data is relatively scarce (e.g., in healthcare). In such cases, explanations can help overcome the obstacles to building highly accurate models, and decision-makers can still use algorithmic decisions effectively.

Third, given their critical and sensitive nature, decisions are not likely to be fully automated soon in areas like healthcare, and machine predictions should be used cautiously. Our results in this study confirm the existence of the biasing effect in that the overall performance of human-AI collaboration strongly depends on the accuracy of the algorithms when AI predictions are presented.

Fourth, another important consideration is the role of participants' prior medical knowledge in shaping their comprehension of both radiology report content and AI-generated explanations. Although we included a self-reported measure of healthcare literacy to approximate domain expertise, this measure provides only a general sense of participants' familiarity with medical concepts. Individuals with more extensive medical knowledge may interpret technical terminology or explanation cues differently, potentially moderating the impact of AI support on decision outcomes. Conversely, those with less familiarity may benefit more from explanatory cues that simplify or contextualize medical language. Future research should explore how varying levels of domain-specific knowledge influence human-AI collaboration, particularly in high-stakes domains like healthcare, and consider incorporating more granular measures of expertise to better tailor AI explanations to user needs.

Fifth, although our study focused on radiology report excerpts related to anatomical assessments, the mechanisms uncovered—particularly how AI explanations influence self-monitoring and final decision accuracy—may generalize to other clinical contexts. In domains such as pathology, dermatology, or even mental health assessments, clinicians and patients routinely interpret textual or semi-structured diagnostic data. In these settings, explanatory cues that highlight salient features or align with human reasoning may similarly support cognitive calibration and improve judgment. Future studies should explore the transferability of these effects across modalities and specialties to assess how explanation design can be adapted to domain-specific workflows and reasoning styles.

Sixth, our study also invites comparison to research on second opinions provided by human professionals. Similar to human collaborators, AI systems offering a second opinion can influence decisions through both the content of the recommendation and the presence of a rationale or explanation. However, unlike human experts, AI systems often lack social cues, experiential context, and perceived accountability—all of which can shape how advice is received and evaluated. Prior research in team cognition (e.g., [82,106]) has shown that explanations from human peers can enhance advice acceptance by promoting understanding and credibility. In clinical settings, second opinions are valued not only for diagnostic correction but also for building confidence and reducing decisional conflict [107]. AI explanations, while more mechanical, may similarly scaffold user reasoning by offering interpretable anchors. Our results suggest that explanations from AI can trigger self-monitoring and override intuitive errors, much like a persuasive human second opinion might. However, the path to trust and reliance likely differs due to the perceived agency, intent, and contextual awareness that human collaborators bring. Future research should explore whether users internalize and weigh second opinions from AI systems differently than those from human experts, particularly in high-stakes domains such as healthcare.

Finally, systems could be designed to elicit users' initial judgment and confidence before presenting AI recommendations—encouraging self-reflection and enabling tailored presentation of advice. This aligns with our finding that explanations are more effective when initial self-confidence is low. Additionally, tools could incorporate a "devil's advocate" feature that presents counterfactual explanations challenging the user's initial framing—even when the AI prediction aligns with

the user's intuition. Doing so could activate metacognitive conflict and help users re-evaluate their reasoning, similar to how human collaborators sometimes challenge each other's assumptions. Furthermore, dynamically adapting explanation strength or framing based on user confidence and prediction confidence could support more calibrated reliance. We encourage designers to view explanations not just as static justifications, but as interactive mechanisms for cognitive support—capable of modulating trust, effort, and decision quality.

## 5.3. Limitations and future research

This study has some limitations and creates opportunities for further research. First, we study this only in the context of annotating medical records. This unique nature of textual data allowed us to develop simple insight mechanisms for the decision-maker. Future research should examine whether these effects are observable using more generalizable tasks. Second, our explanations were created by isolating the features with the highest contributions [108], which is a simple —but not necessarily the most optimal—way of creating explanations. Explanations created by other methods, such as model approximation (e.g., [43]) can also be investigated for their contributions to human judgment accuracy. Furthermore, while our study used feature-based explanations generated from an SVM trained on radiology report sentences, it is important to recognize that these explanations are synthetic in nature. That is, they were created specifically for this experiment using controlled, transparent mechanisms (e.g., ranked unigrams and bigrams). In contrast, real-world AI systems such as ChatGPT or GlassBox models often produce explanations that are generated through more complex, probabilistic, and potentially less transparent methods. These real-world systems may involve dialogue-based, multi-modal, or dynamically updated explanations shaped by conversational context. As such, the effects of explanations observed in our study may not fully generalize to real-world deployments where explanations are noisier, more ambiguous, or inconsistently interpreted. Future research should explore whether the self-monitoring benefits observed here persist under more realistic, end-to-end AI explanation interfaces. Third, while explanations significantly improve the overall accuracy, improvements beyond machine learning prediction accuracy can be explored. For example, in our case, the final judgment is ultimately performed by human judges. If we can identify the characteristics of decision opportunities where humans are most likely to succeed and the situations where machine predictions are most likely to fail, then we can design intelligent decision workflows that delegate decisions to agents who are most likely to succeed. Fourth, we approximate the initial framing measures based on the consensus in the control group. The initial framing can be more accurately measured in a with-in-subject experiment setting by asking participants to complete the task first without AI recommendations before completing the same task again with AI recommendations. However, with this approach, the final task outcome could be influenced by the carryover [109] or the priming [110] effects of the first step. Therefore, we used the responses from the control condition where AI recommendations were not presented. While we believe this is a good proxy for what the initial framings would be for the same task when AI recommendations are present, future studies can focus on extracting initial framing instead of using a proxy. Moreover, as we discussed earlier, explanations have the potential to improve final judgment accuracy. Therefore, merely providing practical explanations can alleviate the need for more complex systems or bigger training datasets. At the same time, implementing explanations requires resources. It might be beneficial to look into optimizing resources and modeling accuracy with AI explanations. We would like to note that we measured domain expertise using a self-reported healthcare literacy item, which served as a proxy for familiarity with medical terminology and context. However, we acknowledge that this is a coarse measure that does not account for broader forms of domain-general expertise. Similarly, while we included a self-reported measure of participants' knowledge of AI technology to approximate their familiarity with algorithmic systems, we acknowledge that this is not an ideal proxy for algorithmic literacy. Algorithmic literacy entails an understanding of how AI systems generate predictions, their underlying assumptions, and appropriate interpretations of their outputs. Our measure likely reflects general familiarity or comfort with AI concepts rather than this deeper cognitive skill set. Future studies should consider incorporate more rigorous assessments of participants' medical familiarity and algorithmic literacy to better capture how these factors shape

human-AI collaboration. Additionally, this study used lay participants recruited from Amazon Mechanical Turk rather than individuals with formal medical training. While this choice may constrain ecological validity for clinical decision-making settings, it reflects a deliberate focus on understanding how general users (e.g., lay patients) interpret AI explanations in the absence of domain expertise. As AI-based tools become increasingly available to patients and the broader public—for example, in direct-to-consumer health platforms or patient-facing EHR systems—understanding how lay individuals rely on AI and engage in self-monitoring is crucial. Finally, our study primarily focuses on participants' initial framing, and their deliberate consideration of AI recommendations when presented. While we control for participants' trust in our work, there is a growing emphasis on the importance of calibrating trust (i.e., avoiding over- or under-reliance [111]) to enhance AI-assisted decision-making [112]. We expect further studies to consider the nuanced differences between trust and reliance [113,114] in studying the effects of explanations on metacognitive processes of AI-assisted decision-making.

## Author contributions

**Conceptualization:** Yegin Genc, Mehmet Eren Ahsen, Zhan Zhang.

**Data curation:** Yegin Genc, Mehmet Eren Ahsen, Zhan Zhang.

**Formal analysis:** Yegin Genc, Mehmet Eren Ahsen.

**Funding acquisition:** Zhan Zhang.

**Investigation:** Yegin Genc, Zhan Zhang.

**Methodology:** Yegin Genc.

**Project administration:** Yegin Genc.

**Resources:** Yegin Genc.

**Writing – original draft:** Yegin Genc.

**Writing – review & editing:** Yegin Genc, Mehmet Eren Ahsen, Zhan Zhang.

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
