## [Decision Letter · Decision Letter 0]

24 Apr 2025

PONE-D-25-11149Investigating the Role of AI Explanations in Lay Individuals’ Comprehension of Radiology Reports: A Metacognition LensePLOS ONE

Dear Dr. Genc,

Thank you for submitting your manuscript to PLOS ONE. After careful consideration, we feel that it has merit but does not fully meet PLOS ONE’s publication criteria as it currently stands. Therefore, we invite you to submit a revised version of the manuscript that addresses the points raised during the review process.

**Major revision:**The paper needs improvments in its technical contents and presentation.Please carefull address the comments by the reviewers in the revision.

We look forward to receiving your revised manuscript.

Kind regards,

Zeyar Aung

Academic Editor

PLOS ONE

3. Please ensure that you have specified a) Did participants provide their written or verbal informed consent to participate in this study?

- In consent please state in Ethics Method section and manuscript if it is written or verbal. If consent was verbal, please explain a) why written consent was not obtained, b) how you documented participant consent, and c) whether the ethics committees/IRB approved this consent procedure.

 [This work was supported in part by funding from the National Institute of Health (Award# 1R15LM014556-01).].

5. We note that you have indicated that there are restrictions to data sharing for this study. PLOS only allows data to be available upon request if there are legal or ethical restrictions on sharing data publicly. For more information on unacceptable data access restrictions, please see http://journals.plos.org/plosone/s/data-availability#loc-unacceptable-data-access-restrictions.

Additional Editor Comments (if provided):

Reviewers' comments:

Reviewer's Responses to Questions

**Comments to the Author**

1. Is the manuscript technically sound, and do the data support the conclusions?

Reviewer #1: Yes

Reviewer #2: Partly

Reviewer #3: Yes

2. Has the statistical analysis been performed appropriately and rigorously? 

Reviewer #1: Yes

Reviewer #2: Yes

Reviewer #3: Yes

3. Have the authors made all data underlying the findings in their manuscript fully available?

Reviewer #1: Yes

Reviewer #2: Yes

Reviewer #3: No

4. Is the manuscript presented in an intelligible fashion and written in standard English?

Reviewer #1: Yes

Reviewer #2: Yes

Reviewer #3: Yes

5. Review Comments to the Author

Reviewer #1: The manuscript presents a timely and well-executed investigation into how AI-generated radiology explanations influence non-experts' understanding of medical images. The authors integrate theories from metacognition to frame their hypothesis and employ a large-scale online experiment with over 4,000 participants. This interdisciplinary approach—bridging XAI, radiology, and cognitive psychology—is commendable and advances our understanding of how explanation formats interact with human reasoning.

The study is technically sound. The hypotheses are clearly stated and rooted in prior theory. The random assignment to explanation formats, the use of real chest X-rays, and the employment of both behavioral and self-reported outcome measures collectively support strong internal validity. The conclusions are supported by appropriate statistical analysis using logistic regression and stratified comparisons.

The manuscript is also notable for its transparency. The dataset, experimental materials, and code are made fully available, adhering to open science best practices. This enhances the credibility and reproducibility of the study.

The writing is clear, and the results are communicated in an accessible way. The framing using metacognitive "direct access" versus "cue-based inference" provides a novel lens on the efficacy of explanations in XAI, and this framing is well-motivated throughout.

I have only a few minor suggestions for the authors:

Consider discussing the potential influence of participants' prior knowledge or medical familiarity on their comprehension.

It would be valuable to elaborate briefly on how the findings might generalize to other modalities or clinical contexts beyond chest X-rays.

A short reflection on the limitations of using synthetic AI explanations (rather than explanations from deployed systems like ChatGPT or GlassBox models) might help ground expectations for real-world translation.

Overall, this is a strong manuscript that makes a valuable contribution to the literature on explainable AI in healthcare. I recommend its acceptance with only minor revisions.

Reviewer #2: In this paper, the authors examine the effects of AI explanations on human decision-making through a metacognitive lens. They conducted a large-scale experiment on Amazon Mechanical Turk, where participants were asked to classify radiology reports as normal or abnormal under three conditions: without AI input (control), with AI prediction only, and with AI prediction plus explanation.

The paper is well-structured and presents some interesting findings, but there are several significant issues that need to be addressed before publication:

- The most critical oversight is the absence of any discussion on "reliance", which is a fundamental concept in the Human-Computer AI Interaction literature. This paper clearly investigates how humans rely on AI with added explanations, yet fails to position itself within this important body of literature. Established measures such as Relative Self-Reliance and Relative AI Reliance could have been adopted to strengthen the analysis.

- The choice of participants raises serious concerns about ecological validity. Using a general audience from Mechanical Turk to perform specialized medical diagnosis tasks is problematic. The authors should justify why they didn't use participants with relevant expertise or at least address this limitation more thoroughly.

- The proxy measures used to categorize participants are questionable. Self-reported healthcare literacy is an overly broad measure that may not accurately reflect domain expertise for the specific task of interpreting radiology reports. Similarly, self-reported confidence has well-documented limitations in the literature that aren't adequately addressed.

- The methodology for creating models with different accuracy levels is poorly explained and justified. The authors trained different models on different training set sizes to simulate varying accuracies, but they don't report the actual accuracy differences between these models. Moreover, using differently sized training sets may result in models that are differently fitted rather than having meaningfully different accuracies.

- "Knowledge of AI technology" is likely not a good indicator of algorithmic literacy, and this limitation is not acknowledged.

- The statistical tests used in the analysis lack justification, and the reporting of significance is incomplete. While the authors claim their hypotheses are supported, they should clarify whether all supported hypotheses have p-values < 0.05. Additionally, the paper would benefit from more detailed reporting of effect sizes to help readers understand the practical significance of the findings.

- The literature review is outdated, with only three papers from 2023 and nothing more recent. This is a rapidly evolving field, and the authors should incorporate more current research.

In conclusion, while the paper addresses an important topic in explainable AI and human-AI collaboration, the methodological issues and gaps in theoretical framing significantly limit its contribution. I recommend major revisions to address these concerns before the paper can be considered for publication.

Reviewer #3: The paper investigates meta-cognitive behaviour of humans working with an AI assistant providing explanations for the purpose of decision making. It specifically focuses on the influence of an AI's recommendation on a human's initial assessment with and without an explanation, when a human's initial assessment was correct or not, and when the human or the AI were confident in their assessments or not. The study was conducted with 4302 participants on Amazon Mechnical Turk.

Strong points:

S1: Important problem with potential impact

S2: Well-conducted and well-described study (although I have a few comments, see W1)

Weak points:

W1: Some comments on the design of the study

W2: Desirable additional comparison

W3: Unclear description of some of the metrics

W4: Desirable advise to designers of AI-assisted decision making tools

I enjoyed reading this paper, not only because it was executed and written well (S2), but also because it studies an important issue (the interaction between human and AI in joint decision making) and the focus of the study is rather intriguing: meta-cognitive processes (S1). Being an AI expert and not a social scientist, I welcome the aid of social scientists with works like these for the purpose of improving the design of AI-support in decision support systems.

W1: I wondered about a few things when analysing your research design. First, isn't it the case that people of varying intelligence may have different meta-cognitive processes? I would have expected some aspect(s) associated with intelligence or level of schooling to be gathered from the participants (table 2) and analysed. Second, the participants form a population that is quite different from the population of healthcare professionals. One should be very careful with extrapolating conclusions to decision making by healthcare professionals. Thirdly, you formulated hypotheses around initial framing, but there is another important case of framing: an apriori assumption that the AI will be correct. I have witnessed this phenomenon with healthcare professionals placing overly strong trust in AI recommendations. Fourthly, in section 3.4.5 you discuss the control variables among which "domain expertise". I find this a bit too crude: a professional with expertise in decision making from another domain is very likely to exhibit similar meta-cognitive behaviour while in decision making in their own domain or in health.

I don't expect that you can fix all the points I raised, nor that all of them are valid, but if you agree with one or more of my comments, then I would welcome them mentioned as a limitation of the study.

W2: In some ways, working with an AI providing a second opinion is different than working with a human providing a second opinion ... but in other ways, they are not so different. I assume there are studies about effects on decision making when receiving a second opinion from another professional with or without an explanation. I would welcome a short discussion in this paper in what way your results show differences or commonalities between teaming up with an AI or teaming up with a human professional in a similar way.

W3: In Section 3.2.1 you say that you use the expert labels of the 276 sentences as golden standard to evaluate the AI predictions and participants' decision outcomes. Section 3.4.1 describes these accuracy metrics, but I fail to recognise the use of the golden standard in the explanation of the metrics. It only speaks of majority vote. Please clarify this apparent inconsistency. Furthermore, why not provide a simple formula for the metrics? It is much more exact and easier to interpret than a description of a formula in natural language.

In Section 3.4.4 you describe how you measure the self-confidence. If I understand correctly, participants are assigned to a group, so a participant in the "with explanations" group will never see sentences under the control condition. So, how do you determine their self-confidence? Furthermore, do I understand correctly that the self-confidence is per candidate and not per sentence (like the AI's confidence)? Please clarify these aspects.

W4: One form of potential practical impact of your work is towards designers of support tools for decision making. Your research may help in making certain design choices. For example, would it make sense to let the AI explicitly play devil's advocate in certain situations, i.e., contrary to its own assessment, provide an explanation contrary to the clinician's own initial framing? Would it make sense to ask a clinician about their initial assessment and confidence before presenting the AI's second opinion and explanation? Perhaps you can think one step further about your results and conclusions towards design recommendations and add a paragraph in the discussion with some thoughts and recommendations directed towards such tool designers.

All in all, I found the paper intriguing to read, interesting results, and it made me to think more about these effects; all characteristics of a paper that deserves to be published. I hope that my attention to explaining some of these thoughts can help strengthening the paper.

Minor remarks:

* Footnote 3: you say that you consider explanation efficiencies out of the scope of the paper. I would expect that a participant being confronted with a conflicting prediction is likely to study the reasoning in the explanation, checking it for being reasonable. So, isn't explanation efficiency inherently included in the scope of the paper?

* Figure 2: too low resolution in the figure. The texts are hardly readable.

* In 3.2.1 you refer to "two AI models". At that point in the paper, the reader doesn't know yet what two AI models you are referring to. A reader learns about it in 3.4.5. Perhaps better to explain the two AI models already in 3.2.1.

* In section 3.4.5 you present two models. It would be handy to present the accuracies of both models as measured with the labels of the golden standard (276 sentences with labels assigned by experts).

* In the conclusion, you mention that there were cases where both AI and human expert were incorrect, but that the explanations triggered a self-monitoring process and led to correct decisions. It would be nice to have a few examples of these in the paper.

6. PLOS authors have the option to publish the peer review history of their article (what does this mean?). If published, this will include your full peer review and any attached files.

Reviewer #1: No

Reviewer #2: **Yes: **Tommaso Turchi

Reviewer #3: **Yes: **Maurice van Keulen

---

## [Author Response · Author response to Decision Letter 1]

13 Jun 2025

Thank you for your feedback

- We uploaded the final version without tracked changes as the main article. We believe this covers issues #1 and #3

- We included a new cover letter with this submission that has our Full Funding statement. We hope you can update online funding form accordingly. We hope this covers issue #2

- We included this in our participation and online consent detail in the beginnig of the methods section as a foot note to the online experiment phrase. We believe this covers issue #4

Thank you again for your consideration

Regards,

Yegin Genc

We've checked your submission and before we can proceed, we need you to address the following issues:

1. We notice that your revision was submitted on [Jun 9 2025], but the manuscript file in your submission's file inventory was uploaded on [Mar 4 2025]. Please upload the latest version of your revised manuscript as the main article file, with the item type 'Manuscript,' ensuring that it does not contain any tracked changes or highlighting. This will be used in the production process if your manuscript is accepted.

[This work was supported in part by funding from the National Institute of Health (Award# 1R15LM014556-01).].

Please provide an amended statement that declares *all* the funding or sources of support (whether external or internal to your organization) received during this study, as detailed online in our guide for authors at https://nam12.safelinks.protection.outlook.com/?url=http%3A%2F%2Fjournals.plos.org%2Fplosone%2Fs%2Fsubmit-now&data=05%7C02%7Cygenc%40pace.edu%7C8cdf8777f36f4ef252a008dda8934a93%7C0799c53eca9a49e88901064a6412a41d%7C0%7C0%7C638852072761260532%7CUnknown%7CTWFpbGZsb3d8eyJFbXB0eU1hcGkiOnRydWUsIlYiOiIwLjAuMDAwMCIsIlAiOiJXaW4zMiIsIkFOIjoiTWFpbCIsIldUIjoyfQ%3D%3D%7C0%7C%7C%7C&sdata=WepcB3rT%2BUaB93OIT7LKTX7Mn1GLL0lH6ggQ7TjeH%2B4%3D&reserved=0. Please also include the statement “There was no additional external funding received for this study.” in your updated Funding Statement.

3. Can you please upload an additional copy of your revised manuscript that does not contain any tracked changes or highlighting as your main article file. This will be used in the production process if your manuscript is accepted. Please amend the file type for the file showing your changes to Revised Manuscript w/tracked changes. Please follow this link for more information: https://nam12.safelinks.protection.outlook.com/?url=http%3A%2F%2Fblogs.plos.org%2Feveryone%2F2011%2F05%2F10%2Fhow-to-submit-your-revised-manuscript%2F&data=05%7C02%7Cygenc%40pace.edu%7C8cdf8777f36f4ef252a008dda8934a93%7C0799c53eca9a49e88901064a6412a41d%7C0%7C0%7C638852072761316957%7CUnknown%7CTWFpbGZsb3d8eyJFbXB0eU1hcGkiOnRydWUsIlYiOiIwLjAuMDAwMCIsIlAiOiJXaW4zMiIsIkFOIjoiTWFpbCIsIldUIjoyfQ%3D%3D%7C0%7C%7C%7C&sdata=bxfadTmj37%2F5I2Cxdyd3dp92y6DKuVOzHmju%2FAuRFIs%3D&reserved=0. Kindly check the yellow highlight on page 5.

4. Thank you for including your ethics statement on the online submission form: "Human Participants. Pace Univerity Institutional Review Board approved the study. The IRB Code # is 19-07.

Participant recruitment and data collection took place between 04/15/2019 and 05/24/2019. In the beginning, participants were presented with an online consent form and provided consent by clicking on a button that reads "I certify that I read and understand the informed consent". Their consent is recorded in the dataset. Upon giving consent, participants complete a demographics questionnaire on a survey-like interface. (this paragraph is copied from the manuscript, section 3.3: Participants and Data Collection Process, paragraph 2) ". To help ensure that the wording of your manuscript is suitable for publication, would you please also add this statement at the beginning of the Methods section of your manuscript file.

---

## [Decision Letter · Decision Letter 1]

5 Aug 2025

Investigating the Role of AI Explanations in Lay Individuals’ Comprehension of Radiology Reports: A Metacognition Lens

PONE-D-25-11149R1

Dear Dr. Genc,

We’re pleased to inform you that your manuscript has been judged scientifically suitable for publication and will be formally accepted for publication once it meets all outstanding technical requirements.

Kind regards,

Zeyar Aung

Academic Editor

PLOS ONE

Additional Editor Comments (optional):

Reviewers' comments:

Reviewer's Responses to Questions

**Comments to the Author**

1. If the authors have adequately addressed your comments raised in a previous round of review and you feel that this manuscript is now acceptable for publication, you may indicate that here to bypass the “Comments to the Author” section, enter your conflict of interest statement in the “Confidential to Editor” section, and submit your "Accept" recommendation.

Reviewer #1: All comments have been addressed

Reviewer #2: All comments have been addressed

2. Is the manuscript technically sound, and do the data support the conclusions?

Reviewer #1: Yes

Reviewer #2: Yes

3. Has the statistical analysis been performed appropriately and rigorously? 

Reviewer #1: Yes

Reviewer #2: Yes

4. Have the authors made all data underlying the findings in their manuscript fully available?

Reviewer #1: Yes

Reviewer #2: Yes

5. Is the manuscript presented in an intelligible fashion and written in standard English?

Reviewer #1: Yes

Reviewer #2: Yes

6. Review Comments to the Author

Reviewer #1: The authors have made commendable revisions that address the concerns raised in the previous round. The inclusion of detailed statistical reporting, clearer clarification of the methodology (particularly around gold-standard labels and confidence metrics), and the thoughtful addition of practical implications for AI tool design have significantly strengthened the manuscript.

The limitations regarding participant expertise and the use of self-reported measures are now appropriately acknowledged, and the manuscript provides a balanced discussion around generalizability and interpretive caution. The writing is clear, and the interdisciplinary framing continues to be a notable strength.

While the manuscript would have further benefited from engaging with the Human-AI reliance literature, the current version remains technically sound, methodologically transparent, and a valuable contribution to the field of explainable AI in healthcare contexts.

I recommend the manuscript for publication.

Reviewer #2: I'm satisfied with how the authors addressed my comments and other reviewers', I recommend accepting the paper in its current form.

7. PLOS authors have the option to publish the peer review history of their article (what does this mean?). If published, this will include your full peer review and any attached files.

Reviewer #1: No

Reviewer #2: **Yes: **Tommaso Turchi

---

## [Editor Report · Acceptance letter]

PONE-D-25-11149R1

PLOS ONE

Dear Dr. Genc,

I'm pleased to inform you that your manuscript has been deemed suitable for publication in PLOS ONE. Congratulations! Your manuscript is now being handed over to our production team.

Kind regards,

on behalf of

Dr. Zeyar Aung

Academic Editor

PLOS ONE